# Increased Resection at DSBs in G_2_-Phase Is a Unique Phenotype Associated with DNA-PKcs Defects That Is Not Shared by Other Factors of c-NHEJ

**DOI:** 10.3390/cells11132099

**Published:** 2022-07-02

**Authors:** Huaping Xiao, Fanghua Li, Emil Mladenov, Aashish Soni, Veronika Mladenova, Bing Pan, Rositsa Dueva, Martin Stuschke, Beate Timmermann, George Iliakis

**Affiliations:** 1Institute of Medical Radiation Biology, University Hospital Essen, University of Duisburg-Essen, 45147 Essen, Germany; huaping.xiao@uk-essen.de (H.X.); fanghua.li@uk-essen.de (F.L.); emil.mladenov@uk-essen.de (E.M.); aashish.soni@uk-essen.de (A.S.); veronika.mladenova@uk-essen.de (V.M.); bing.pan@uk-essen.de (B.P.); rositsa.dueva@uk-essen.de (R.D.); 2Division of Experimental Radiation Biology, Department of Radiation Therapy, University Hospital Essen, University of Duisburg-Essen, 45147 Essen, Germany; martin.stuschke@uk-essen.de; 3Department of Particle Therapy, University Hospital Essen, West German Proton Therapy Centre Essen (WPE), West German Cancer Center (WTZ), German Cancer Consortium (DKTK), 45147 Essen, Germany; beate.timmermann@uk-essen.de; 4Institute of Physiology, University Hospital Essen, University of Duisburg-Essen, 45147 Essen, Germany; 5German Cancer Consortium (DKTK), Partner Site University Hospital Essen, German Cancer Research Center (DKFZ), 45147 Essen, Germany

**Keywords:** DSB repair, DNA-PKcs, c-NHEJ, DNA end-resection, ionizing radiation

## Abstract

The load of DNA double-strand breaks (DSBs) induced in the genome of higher eukaryotes by different doses of ionizing radiation (IR) is a key determinant of DSB repair pathway choice, with homologous recombination (HR) and ATR substantially gaining ground at doses below 0.5 Gy. Increased resection and HR engagement with decreasing DSB-load generate a conundrum in a classical non-homologous end-joining (c-NHEJ)-dominated cell and suggest a mechanism adaptively facilitating resection. We report that ablation of DNA-PKcs causes hyper-resection, implicating DNA-PK in the underpinning mechanism. However, hyper-resection in *DNA-PKcs*-deficient cells can also be an indirect consequence of their c-NHEJ defect. Here, we report that all tested *DNA-PKcs* mutants show hyper-resection, while mutants with defects in all other factors of c-NHEJ fail to do so. This result rules out the model of c-NHEJ versus HR competition and the passive shift from c-NHEJ to HR as the causes of the increased resection and suggests the integration of DNA-PKcs into resection regulation. We develop a model, compatible with the results of others, which integrates DNA-PKcs into resection regulation and HR for a subset of DSBs. For these DSBs, we propose that the kinase remains at the break site, rather than the commonly assumed autophosphorylation-mediated removal from DNA ends.

## 1. Introduction

The risks posed by DNA double-strand breaks (DSBs) to the genomic integrity of higher eukaryotes are mitigated by a network of signaling pathways collectively termed the DNA damage response (DDR). DDR detects DSBs and coordinates a wide spectrum of cellular responses, including checkpoint activation and DSB repair [1,2]. Notably, the biological toxicity of DSBs derives to a great extent from the ways they are processed, rather than from lack of processing altogether [3,4,5]. Indeed, although four mechanistically distinct repair pathways, classical non-homologous end-joining (c-NHEJ), homologous recombination (HR), alternative end-joining (alt-EJ) and single-strand annealing (SSA), remove DSBs from the genome, only HR ensures sequence restoration at the break-site and reunion of the original DNA ends. c-NHEJ, alt-EJ and SSA, all can restore structural integrity in the genome, but are associated with mutations, deletions or insertions at the junction, and can even misjoin ends to form chromosomal translocations [3,4,5,6]. Thus, counterintuitively, the genomic instability that originates from DSBs is directly linked to the repair pathways that evolved to repair them.

It is relevant that in lower eukaryotes, such as the yeast (and bacteria), DSB repair is mainly the task of HR, with the remaining repair pathways having rather secondary roles. HR is coordinated in the yeast by a DDR network organized around the ataxia telangiectasia-mutated and Rad3-related (ATR) homolog, Mec1, and to a lesser extent, by the ataxia telangiectasia mutated (ATM) homolog, Tel1 [7,8]. In contrast to the dominance of HR in the yeast, in higher eukaryotes, c-NHEJ dominates [6,8,9]. This shift to c-NHEJ coincides with the evolutionary appearance of the DNA-dependent protein kinase catalytic subunit (DNA-PKcs) [10] and is accompanied by a shift in DDR-organization towards ATM while maintaining contributions from ATR [11,12,13,14,15]. In this operational framework of DDR, DSBs are seemingly directed by design to a fast operating, error-prone repair pathway and error-free HR only repairs small subsets of DSBs, when c-NHEJ fails [16,17,18].

We reasoned that a fixed evolutionary hardwiring of DSB-processing to c-NHEJ is difficult to reconcile with the high stability requirements of higher eukaryotic genomes and explored whether mechanisms or conditions increasing HR utilization exist but have hitherto been missed. Indeed, a systematic analysis of the repair of ionizing radiation (IR) induced DSBs in G_2_-phase human and rodent cells, where all repair pathways are fully active, revealed that the engagement of HR strongly depends on the IR dose administered, i.e., on the load of DSBs induced in the genome. Specifically, we discovered that HR can repair ~50% of DSBs at doses below 0.5 Gy, but that it only repairs ~10% at 2 Gy and that its contribution is actually undetectable above 5–10 Gy when the genome becomes shuttered and thus, presumably, profoundly destabilized for HR (IR induces ~40 DSBs/Gy in the G_2_-phase human genome) [19,20,21,22]. This observation reveals that higher eukaryotic cells are endowed with mechanisms enabling them to engage HR more frequently than commonly thought [16,18,23], albeit in a strongly DSB-load-dependent manner. Since the inception of HR requires DNA end-processing that involves nucleolytic degradation of the 5′-strands of DNA ends [24,25,26]-known as DNA end-resection (henceforth here simply resection), it follows that increased engagement of HR at low IR-doses will increase the proportion of DSBs that will need to undergo resection.

Analysis of the mechanisms that control the G_2_-checkpoint [27,28] also uncovered a strong dependence on IR-dose and direct links to resection. Specifically, low DSB-loads activate in cells irradiated in G_2_-phase a checkpoint that is fully controlled by ATR, suggesting that only resected DSBs activate the checkpoint [27]. Strikingly, at low DSB-loads, ATR also regulates resection and ATR inhibition suppresses resection, linking thus ATR to the regulation of resection and HR [28,29]. As a direct consequence of these connections, HR mutants or wild-type cells with suppressed HR display, at low IR doses, a strongly compromised G_2_-checkpoint. Collectively, these results define at low DSB loads in higher eukaryotes a condition where DDR and DSB repair assume a yeast-like organization, with enhanced HR utilization and dominant ATR signaling. Strikingly, with increasing DSB-load, the dependence of resection and checkpoint on ATR diminishes and independent contributions from ATM become evident, as c-NHEJ gains ground [27,30].

Increased resection and increased HR engagement with decreasing DSB-load generate a conundrum in a presumably c-NHEJ-dominated cell because DSBs will be promptly bound by the following two highly abundant factors of c-NHEJ: the Ku70/Ku80 heterodimer and DNA-PKcs to form at DNA ends the DNA-PK holoenzyme [6,31,32,33]. While this recruitment is in line with the dominance of c-NHEJ at high IR doses [21,22], it leaves open how HR orchestrates its engagement to ~50% of DSBs at low IR doses. Notably, such engagement requires a mechanism adaptively facilitating resection with diminishing IR dose.

Such a mechanism must somehow involve DNA-PK, as it firmly holds the DSB ends from the moment of their production, and can be mediated either by the widely accepted regulated removal of DNA-PK from the DNA ends [28,32,33,34,35,36,37,38] or alternatively, the here postulated integration of DNA-PK to the resection process. Indeed, systematic analysis of DNA-PKcs mutations affecting the phosphorylation status of the protein generated results compatible with the involvement of DNA-PKcs in the regulation of HR [32,33]. Furthermore, recent biochemical investigations uncover a direct involvement of DNA-PKcs in resection, although the removal of DNA-PK from DNA ends remains a requirement for advanced steps of resection-dependent DSB repair in the models developed by the authors [39,40].

Our own work from the above-outlined studies also suggests an involvement of DNA-PKcs in the G_2_-checkpoint and resection. Thus, in cells irradiated in the G_2_-phase, genetic ablation of DNA-PK causes hyper-resection and checkpoint hyperactivation, manifesting as delayed checkpoint recovery [27,28]. Both functions, if directly dependent on DNA-PK, will require DNA-PK to remain at the DSB site for many hours, and thus, much longer than needed for c-NHEJ.

However, the hyper-resection and checkpoint hyperactivation observed after ablation of DNA-PK can also be interpreted as an indirect consequence of the associated suppression of c-NHEJ, which passively frees up DNA ends for HR. Indeed, it is frequently assumed that the engagement at a DSB of c-NHEJ or HR is decided stochastically—as the outcome of competition. In the face of such competition, c-NHEJ will naturally win [8,41], aided by the high affinity of Ku and DNA-PKcs for DNA ends and their high abundance in cells—particularly human cells [6,42,43,44,45,46]. It then directly follows that genetic ablation of c-NHEJ will benefit HR, which, owing to its slower kinetics, causes persistent checkpoint activation as a default response. It is evident, therefore, that to implicate firmly DNA-PK in the regulation of resection and checkpoint in the above set of experiments, it is essential to rule out a passive response as the underpinning mechanism.

Reasoning based on competition predicts similar phenotypes for resection and checkpoint for mutants of DNA-PKcs and mutants of all other factors of c-NHEJ and offers a means to test its validity. In our previous work, preliminary experiments using c-NHEJ mutants defective in factors other than DNA-PKcs failed to duplicate the results of DNA-PKcs deficient cells [27,28]. This questioned the explanation of indirect DNA-PK involvement and inspired the present work.

Here, we outline experiments demonstrating that all tested DNA-PKcs mutants uniformly show hyper-resection, while none of the mutants defective in other factors of c-NHEJ actually do so. This result rules out the model of the passive shift from c-NHEJ to HR as the cause of increased resection and allows the integration of DNA-PKcs into its regulation.

## 2. Materials and Methods

### 2.1. Cell Culture

Cells were grown at 37 °C in a humidified atmosphere of 5% CO_2_ in air. A549 *DNA-PKcs^−/−^* and parental A549 cells; HCT116 *LIG4^−/−^* and parental HCT116 cells; HeLa *LIG4^−/−^* and parental Hela cells; CHO mutants and their wild-type counterparts were grown in McCoy’s 5A medium. The DNA *LIG4 (LIG4)* deficient, 180BRM; the *XRCC4-like factor (XLF)* deficient, P2 hTert; the *Artemis*-deficient, CJ179 and the normal human fibroblasts, 82-6 hTert were maintained in minimum essential medium (MEM), supplemented with 1% non-essential amino acids (NEAA). *Paralogue of XRCC4 and XLF (PAXX)^−/−^* RPE-1 hTert and parental cells were grown in Dulbecco’s Modified Eagle’s Medium (DMEM). All growth media were supplemented with 10% fetal bovine serum (FBS). All cell lines used in the study were routinely tested for mycoplasma contamination.

### 2.2. Irradiation

Cells were exposed to X-rays at room temperature (RT) using a 320 kV X-ray machine with a 1.65 Al filter (GE Healthcare, Solingen, Germany). The dose rate at 500 and 750 mm distance from the source was 3.5 and 2.1 Gy/min, respectively.

### 2.3. Treatment of Cell Cultures with Inhibitors

DNA2 inhibitor (NIH, Developmental Therapeutics Program, Cat# NSC-105808) was dissolved in dimethylsulfoxid (DMSO) at 10 mM and used at a final concentration of 4 µM. Mirin, an MRE11 inhibitor (2-amino-5-[(4-hydroxyphenyl) methylene]-4(5H)-thiazolone, Santa Cruz Biotechnology) was dissolved in DMSO at 100 mM and was used at 75 µM final concentration. PFM01, a specific inhibitor of MRE11 endonuclease function ((5Z)-5-[(4-Hydroxyphenyl) methylene]-3-(2-methylpropyl)-2-thioxo-4-thiazolidinone, Tocris) was dissolved in DMSO at a 50 mM concentration and was used at a final concentration of 10 μM. In the experiments presented here these two MRE11 inhibitors were used in combination. NU7441 (8-(4-Dibenzothienyl)-2-(4-morpholinyl)-4H-1-benzopyran-4-one, Selleckchem), a specific DNA-PKcs inhibitor was dissolved in DMSO at a final concentration of 10 mM and was used at a working concentration of 10 µM. Unless indicated otherwise, all inhibitors were added to the cell cultures 1 h before irradiation and were maintained until collecting cells for analysis.

### 2.4. RNA Interference

To deplete relevant target proteins in human cells, knockdown experiments were carried out using specific siRNAs against the following proteins of interest: CtBP-Interacting Protein (CtIP), MRE11, RAD50, NBS1, Exonuclease 1 (EXO1), DNA2, Bloom syndrome (BLM), Ku80 and DNA-PKcs using siRNAs as indicated in Appendix A. When more than one siRNA is listed for a specific target protein, they are utilized as a pool. The siRNAs were delivered to the cells by nucleofection using the Nucleofector-2B device (Lonza, Basel, Switzerland). The program X-20 was utilized for siRNA transfection of A549 *wt* and A549 *DNA-PKcs^−/−^* cell lines, while the hamster cells (CHO10B4, CHOK1 and xrs6) were electroporated by using program U-23. The knockdown efficiency in every experiment was assessed by western blot analysis 48 h after nucleofection.

### 2.5. Quantitative Image-Based Cytometry (QIBC) Analysis by Indirect Immunofluorescence (IF)

For indirect IF analysis [19,27], cells were grown on poly-L-lysine (Biochrom) coated coverslips. Thirty minutes before irradiation cells were incubated in media containing 2 μM of 5-ethynyl-2′-deoxyuridin (EdU), which labels S-phase cells. Immediately thereafter, cells were irradiated and EdU was washed out by exchanging growth medium with fresh EdU-free growth medium. At the time of collection (1–6 h), cells were permeabilized in PBS supplemented with 0.25% Triton X-100 (Carl Roth, Karlsruhe, Germany) for 5 min on ice. Subsequently, cells were fixed in PFA-solution (3% paraformaldehyde, 2% sucrose in PBS) for 15 min at RT. After washing with phosphate-buffered saline (PBS), samples were blocked in PBG blocking buffer (0.2% fish-skin gelatin, 0.5% BSA fraction V, in PBS) overnight at 4 °C. Primary antibody against RPA70 (αSSB70B, mouse hybridoma cell line kindly provided by Dr. G. Hurwitz) was diluted (from 1:200 to 1:1000, depending on Ab concentration) in PBG solution and cells were incubated for 1.5 h at RT. The coverslips were washed three times with PBS and cells were incubated in Alexa Fluor 647-conjugated secondary antibody, applied at 1:400 dilution, for 1.5 h at RT. The EdU signal was developed using an EdU staining kit, (Click-It, Thermo Fisher Scientific Waltham, MA, USA), according to the manufacturer’s instructions. Finally, cells were counterstained with 0.2 µg/mL 4′,6-diamidin-2-phenylindol (DAPI, Thermo Fisher Scientific) for 10 min at RT and coverslips were mounted in PromoFluor antifade mounting media (PromoCell, Heidelberg, Germany).

AxioScan.Z1 (Carl Zeiss Microscopy) was utilized to scan selected areas of 4 × 4 mm, containing approximately 10.000–20.000 cells. QIBC analysis combining EdU and DAPI signals allowed us to discriminate the cell cycle phase in which cells were at the time of irradiation. In order to calculate the RPA70 intensity in specifically selected G_2_-phase cells, cellular segmentation analyses were carried out on Imaris 9.5.1 software (Bitplane, Zürich, Switzerland) and the generated data was converted into the proper format for utilization with a flow cytometry software (Kaluza 2.1; Beckman Coulter, Krefeld, Germany). After applying the proper gates, corresponding to the cells of interest, quantitative signal analysis of RPA70 intensity was plotted as histogram plots (Appendix A). Refer to Appendix A for additional details.

### 2.6. Flow Cytometry (FC) Analysis of DNA End-Resection by RPA70 Quantification

For DNA end-resection analysis by FC using RPA70 detection, exponentially growing cells were pulse-labeled for 30 min with 10 µM EdU. After EdU incubation, the growth medium was removed and cells were rinsed once with pre-warmed PBS, returned to growth medium and exposed to X-rays. At different times thereafter, cells were collected by trypsinization and unbound RPA was extracted by incubating pellets for 5 min in ice-cold PBS containing 0.25% Triton X-100. Cells were spun-down for 5 min and pellets were fixed for 15 min using PFA solution. Cells were blocked with PBG blocking buffer overnight at 4 °C and were incubated for 1.5 h with a specific monoclonal antibody raised against RPA70 (see above). Cells were washed twice with PBS and incubated for 1 h with a secondary antibody conjugated with Alexa Fluor 488. Subsequently, EdU signal was developed using an EdU staining kit according to the manufacturer’s instructions. Finally, cells were stained with 40 µg/mL propidium iodide (PI, Sigma-Aldrich, Taufkirchen, Germany.) at RT for 15 min. Three-parameter analysis was carried out in a flow cytometer (Gallios, Beckman Coulter). Similar to IF, EdU-negative G_2_-phase cells are discriminated by their EdU and PI intensity signals. For quantification, the Kaluza 2.1 software was used (Beckman Coulter, Krefeld, Germany). Experiments were replicated 3 times and typically representative histograms from one experiment are shown.

### 2.7. Polyacrylamide Gel Electrophoresis (SDS-PAGE) and Western Blot Analysis

Cells were collected and washed twice in ice-cold PBS. Approximately 5 × 10^6^ cells were lysed for 30 min in 0.4 mL of ice-cold RIPA buffer (Thermo Fisher Scientific) supplemented with Halt^TM^ phosphatase and protease inhibitor cocktails (Thermo Fisher Scientific). Lysates were spun down for 10 min at 11,000× *g* at 4 °C and the protein concentration in the supernatant, containing all soluble proteins, determined using the Bradford assay. Standard protocols for SDS-PAGE and immunoblotting were employed. Unless otherwise indicated, 40 μg of whole-cell extract were loaded on each lane and after electrophoresis, proteins were transferred to a nitrocellulose membrane. The primary antibodies were as follows: anti-Ku80 (Santa Cruz Biotechnology, Santa Cruz, CA, USA sc-9034), anti-DNA-PKcs (Thermo Fisher Scientific Pierce, USA PA1-23197), anti-CtIP (Cell Signaling Technology, Inc, Boston, MA, USA 9201S), anti-EXO1 (GeneTex, Irvine, CA, USA GTX109891), anti-DNA2 (Proteintech, Rosemont, IL, USA 18727-1-AP), anti-BLM (Santa Cruz Biotechnology, USA SC-365753), anti-MRE11 (Novus Biologicals, Minneapolis, MN, USA, NB 100-473), anti-NBS1 (GeneTex, USA, GTX70224), anti-RAD50 (GeneTex, USA, GTX70228), anti-GAPDH (MERCK, Darmstadt, Germany, MAB374), anti-Ku70 (GeneTex, USA, GTX23114); they were used at 1:500–1:5000 dilution. The secondary antibodies were anti-mouse and anti-rabbit IgG conjugated with IRDye680 or anti-rabbit and anti-mouse IgG conjugated with IRDye800 (LI-COR Biosciences, Lincoln, NE, USA, 926-68020, 926-68021, 926-32211 and 926-322210) at 1:10,000 dilution. Immunoblots were visualized by scanning the nitrocellulose membranes on the Odyssey infrared scanner (LI-COR Biosciences). The raw pseudo-colored western blot images were converted to grayscale by Odyssey imaging software. Additionally, western blot images were processed using the brightness and contrast functions incorporated into the Odyssey software.

### 2.8. Cytogenetic Analysis of Translocation Formation in Cells Irradiated in G_2_-Phase

Chromatid translocations were analyzed in irradiated G_2_-phase cells using the protocol described earlier [47,48]. Briefly, exponentially growing cells were exposed to 1 Gy of IR and incubated at 37 °C for 3 h prior to adding Colcemid (0.1 μg/mL, L-6221, Biochrom AG, Berlin, Germany) for 1 h to enrich metaphases. Cells were harvested and processed using standard cytogenetics procedures. Bright-field microscopy (Olympus, Vanox-T, Tokyo, Japan) and a Metasystems station (Altlussheim, Germany) with a microscope (AxioImager.Z2, Zeiss, Oberkochen, Germany) and automated image capture and analysis capabilities were employed for the analysis. Standard criteria were used for scoring chromatid translocations.

### 2.9. Statistical Analysis

The statistical analysis of the results obtained as part of this work was carried out on SigmaPlot v14 using t-test to calculate the *p*-values. In addition, the online version of the “comparison of means” calculator on MedCalc webpage (https://www.medcalc.org/calc/comparison_of_means.php) was utilized. Accessed on 10–17 April 2022. The resulted significance values (*p*-values) calculated from experiments shown in the main and Appendix A are summarized in Appendix A. 

## 3. Results

### 3.1. Specific Analysis of Resection in G_2_-Phase Cells That Are also Irradiated in G_2_-Phase

We have reported that the wiring between DNA-PKcs, ATM and ATR in the regulation of checkpoint and resection in the G_2_-phase changes profoundly between cells irradiated in the G_2_-phase and cells irradiated in the S-phase that later enter the G_2_-phase [27,28]. Specifically, cells exposed to low IR doses in G_2_-phase show responses that are epistatically regulated by ATM and ATR, with DNA-PKcs facilitating checkpoint recovery and preventing hyper-resection. On the other hand, in cells irradiated in S-phase, G_2_-checkpoint, but not resection, is regulated by ATR, while DNA-PKcs and ATM couple and assume similar functions, supporting checkpoint recovery and preventing hyper-resection. These profound cell cycle-dependent regulatory adaptations of DDR necessitated in the present study the design of experiments analyzing resection in the G_2_-phase, specifically in cells also irradiated in the G_2_-phase, avoiding thus confounding interferences from the S-phase cells entering the G_2_-phase.

We have previously reported a QIBC-based approach that achieves this goal and is applicable at relatively low doses [19,27,28]. Thereby, cells are incubated with EdU for 30 min just prior to IR exposure, to label cells in S-phase at the time of irradiation. Subsequently, resection is analyzed at different times after IR, using immunostaining to quantify chromatin-bound RPA70 signal—a widely accepted measure of resection [49,50]. The analysis is restricted to the G_2_-phase cells identified as such by their DAPI-signal intensity and is further restricted to cells irradiated in the G_2_-phase by exploiting their EdU negative status (EdU^−^). Cells in the G_2_-phase at a specific time after IR that are positive for EdU (EdU^+^) are cells irradiated in the S-phase that have entered the G_2_-phase and are thus excluded from the analysis. Appendix A outline for A549 and 82-6 hTert cells, respectively, QIBC results of a typical experiment and shows the gates adopted to measure RPA signal and thus resection in the EdU^−^ G_2_-phase cells. The robust RPA signal increase over that of unirradiated cells shown in Appendix A after exposure to 2 and 4 Gy validates the method for resection analysis at low IR doses.

Similarly-treated cell populations can also be processed by FC to analyze resection after exposure to IR doses above 5 Gy. Appendix A show data from a typical experiment using A549 and 82-6 hTert cells, respectively. Here again, the robust RPA signal increase over unirradiated controls measured after exposure to 10 Gy in Appendix A validates the method for the analysis of resection in the G_2_-phase irradiated cells.

### 3.2. Genetic Ablation of DNA-PKcs Causes Hyper-Resection in Cells Irradiated in G_2_-Phase

We previously reported that after IR, the DNA-PKcs mutants M059J and HCT116 *DNA-PKcs^−/−^* show hyper-resection when analyzed in G_2_-phase [27]. To investigate further the generality of this observation, we generated a *DNA-PKcs^−/−^* cell line by using CRISPR/Cas9 technology on the A549 cell background and carried out similar experiments. Figure 1A shows DNA-PKcs expression in the selected clone and parental cells that document the desired *DNA-PKcs* deficiency.

The QIBC results in Figure 1B,C show hyper-resection in the DNA-PKcs deficient clone, as compared to parental cells, 3 h after exposure to 2 or 4 Gy. Analysis of similarly treated cells by FC at different times after exposure to 10 Gy (Figure 1D,E) confirms the effect at higher doses. Thus, in addition to the M059J/M059K and HCT116 *wt/DNA-PKcs^−/−^* pairs, A549 cells also show hyper-resection at DSBs in G_2_-phase following ablation of DNA-PKcs.

We inquired into the species specificity of this effect and focused on rodent Chinese hamster ovary (CHO) cells for the following two reasons: First, a number of widely used c-NHEJ mutants, including several *DNA-PKcs* deficient mutants, have been generated on this genetic background. Second, the levels of DNA-PKcs are in rodent cells considerably lower than in human cells (Figure 2A), and it is conceivable that expression differences of this magnitude may have regulatory consequences for the resection phenotype.

The QIBC results summarized in Figure 2B,C show that three of the most widely used *DNA-PKcs* CHO mutants, V3, irs20 and XR-C1-3, resect 1 h after exposure to 2 or 4 Gy more extensively than wild-type cells. Increased levels of resection are also observed by FC, 1 h after exposure to 10 Gy (Figure 2D,E). Collectively, these results show that ablation of DNA-PKcs is associated with hyper-resection in the G_2_-phase, not only across different mutants of the same species but also across species. It follows that in *DNA-PKcs* proficient cells, the kinase functions to somehow curtail resection.

### 3.3. Inhibition of DNA-PKcs Using Small Molecule Inhibitors Fails to Cause Hyper-Resection

To investigate the role of DNA-PKcs activity in the resection phenotype outlined above, we carried out similar experiments using wild-type cells after treatment with the DNA-PKcs specific inhibitor, NU7441 (DNA-PKcsi). Strikingly, the QIBC results in Appendix A show that in 82-6 hTert cells exposed to 1 or 4 Gy, specific inhibition of kinase activity fails to generate 3 h later the hyper-resection phenotype documented in all DNA-PKcs mutants tested, and actually causes a small but significant decrease in resection after exposure to 1 Gy. Moreover, A549 and M059K cells exposed to 10 Gy and analyzed by FC 3 h later fail to show hyper-resection after treatment with the DNA-PKcs inhibitor (Appendix A). Because DNA-PKcs inhibition and the enzyme null condition typically present in mutants have been reported to generate different effects on various endpoints, and because inhibited DNA-PKcs is thought to occasionally exert dominant-negative effects by locking the enzyme on DSB ends [8,40,51], we infer similar mechanistic peculiarities in the regulation of the hyper-resection phenotype studied here.

### 3.4. The Curtailing Function of DNA-PKcs in Resection Is Independent of Ku

To investigate whether the function of DNA-PKcs in regulating resection requires Ku, we carried out resection analysis in xrs6, a *Ku80* mutant of CHO cells. The QIBC results in Figure 3A,B show that 3 h after exposure to 2 or 4 Gy, resection levels are similar in the xrs6 mutant and parental CHO cells. A similar analysis of resection by FC, 1 h after exposure to 10 Gy, confirms comparable levels of resection in the two cell lines (Figure 3C,D). Furthermore, two *wt* CHO cell lines (K1 and 10B4) show after knockdown of Ku80 (Appendix A) unchanged resection levels (Appendix A), 3 h after exposure to 4 Gy and analysis by QIBC. Indeed, resection in this set of samples is similar to xrs6 cells (Appendix A) that were included as a positive control. A similar analysis by FC confirms these observations 1 h after exposure to 10 Gy (Appendix A).

To test whether the effect of DNA-PKcs on resection is Ku-independent also in cells with high DNA-PKcs levels, we analyzed resection in A549 cells after knockdown of Ku80. Figure 3E shows the knockdown of Ku80, 24 h after transfection with the corresponding siRNA. Notably, despite Ku80 knockdown, resection remains unaffected in cells exposed to 10 Gy and analyzed 1–6 h later by FC (Figure 3F,G). This surprising outcome is similar to DNA-PKcs inhibition results and suggests that the curtailing effect of DNA-PKcs is sensitive to inhibitor-mediated kinase inhibition and does not require Ku.

### 3.5. Defects in Factors of c-NHEJ, Other Than DNA-PKcs, Fail to Cause Hyper-Resection at DSBs

As highlighted in the Introduction, the simplest interpretation of the hyper-resection measured in DNA-PKcs deficient cells is as an indirect consequence of the associated c-NHEJ defect. Although the above results with *Ku* mutants and DNA-PKcs inhibitors, as well as our earlier work [27], failed to provide support for this model, we wished to further consolidate these findings and carried out more experiments with mutants of key c-NHEJ factors.

LIG4 is a key component of c-NHEJ, with deficiencies causing some of the strongest defects in DSB processing as compared to other mutants. As a direct consequence, LIG4 defects are expected to free up the maximum number of DSBs for resection-dependent processing. Therefore, we investigated how ablation of LIG4 in three well-established model systems affects resection in the G_2_-phase. Specifically, we compared the 180BRM *LIG4*-deficient human fibroblasts with 82-6 hTert normal human fibroblasts. We also compared HCT116 *LIG4^−/−^* cells with parental HCT116 cells and a *LIG4^−/−^* mutant generated by gene knockout in HeLa cells with the parental cell line. The QIBC results in Figure 4A show that resection in *LIG4*-deficient HCT116 cells 3 h after 2 or 4 Gy is similar to that in their wild-type counterparts. Similar results are also obtained in *LIG4* deficient 180BR-IM and HeLa cells (Appendix A, respectively). Analysis of resection using FC in the same *LIG4* deficient cell lines up to 6 h after exposure to 10 Gy confirms this observation (Appendix A).

XRCC4, XLF and PAXX [6,52] are critical LIG4 co-factors and defects are associated with suppression of LIG4 activity and thus of c-NHEJ. QIBC analysis summarized in Figure 4B,C shows that in P2 hTert cells, an *XLF* deficient human fibroblast cell line and in RPE-1 hTert *PAXX^−/−^* cells, a mutant generated by CRISPR/Cas9 mediated gene knockout in RPE-1 cells, resection develops at levels indistinguishable from those of 82-6 hTert and the parental RPE-1 hTert cells, respectively. A similar analysis by FC confirms this observation in the same mutants after exposure to 10 Gy and analysis 1–6 h later (Appendix A), and extends the observation to XR-1 cells—an *XRCC4* CHO mutant (Appendix A).

Artemis is an endo/exo-nuclease with well-defined roles in the DNA end-processing during V(D)J recombination, but less well-defined contributions to end processing during c-NHEJ [53]. *Artemis* defects are associated with relatively small effects on DSB rejoining by c-NHEJ—typically of a magnitude similar to that observed in *ATM*-deficient cells [54]. The QIBC and FC results summarized in Appendix A and Appendix A, respectively, show that *Artemis* defects in CJ179 hTert human fibroblasts fail to increase resection in G_2_-phase after exposure to 2 and 4 Gy and analysis by QIBC 3 h later, or exposure to 10 Gy and analysis by FC 3 h later.

Collectively, the above results demonstrate a unique effect of DNA-PKcs ablation in resection in the G_2_-phase that is not shared by any other factors of c-NHEJ, including Ku and the inhibited DNA-PK holoenzyme. In this presumably non-canonical function, DNA-PKcs likely remains at DSBs for hours, rather than the minutes they normally spend at DSBs during c-NHEJ.

### 3.6. Hyper-Resection in DNA-PKcs Mutants Causes an Explosion in Chromosomal Translocation Formation

Seeking independent support for the enhanced resection observed in *DNA-PKcs* mutants, we analyzed translocation formation under analogous conditions of cell irradiation. In these experiments, we exposed wild-type CHO cells and different c-NHEJ mutants to 1 Gy of X-rays and scored translocation formation by analyzing metaphases collected at 4 h after IR and by adding colcemid 3 h after IR. Since only cells irradiated in the G_2_-phase reach metaphase within this time, the results reflect responses of cells irradiated in the G_2_-phase—as was the case in the analysis of resection. Figure 4D shows typical metaphases from irradiated CHO10B4 cells as well as the c-NHEJ mutants (V3, XR-1 and xrs6). Strikingly, while translocation formation increases modestly (up to twofold), as expected, in the XR-1 and xrs6 mutants (Figure 4E), V3 cells show an explosion in translocation formation with an over five-fold increase over CHO10B4 cells. Since several studies link resection to translocation formation [47,48,55,56,57], we interpret the result as additional evidence that ablation of DNA-PKcs increases resection.

### 3.7. Factors Involved in DNA End Resection in the Absence of DNA-PKcs

We explored whether the hyper-resection detected in DNA-PKcs deficient cells is mediated by the same set of factors catalyzing resection in wild-type G_2_-phase cells. Since CtIP is a key component of the resection apparatus [58,59], we carried out knockdown experiments in A549 cells and compared the results to those of their *DNA-PKcs^−/−^* counterparts. Figure 5A confirms efficient knockdown in both the parental cell line and mutant. Notably, CtIP knockdown causes, in both cell lines, a nearly complete suppression of resection, manifesting 3 h after exposure to 10 Gy (Figure 5B,C).

Since CtIP initiates resection together with the MRN complex, we explored how the depletion of MRN components affects this response. Appendix A shows extensive but incomplete depletion of MRE11, RAD50 and NBS1 in both cell lines, which only modestly reduces resection at 3 h after exposure to 10 Gy (Appendix A). Since incomplete knockdown may be the reason for the incomplete suppression of resection, we carried out complementary studies using inhibitors.

PFM01 is a specific inhibitor targeting the MRE11-associated endonuclease activity [60], while Mirin is an inhibitor of the MRE11-associated exonuclease activity. Owing to this complementarity of activity, we used the inhibitors in combination. Strikingly, incubation of cells with this inhibitor-cocktail completely suppresses resection—both in *DNA-PKcs* proficient cells as well as in *DNA-PKcs* deficient cells (Figure 5D,E). Collectively, the results demonstrate that the canonical mechanism of resection initiation through the activities of CtIP and MRN remains active in *DNA-PKcs* deficient cells and underpins the hyper-resection phenotype.

Having established canonical initiation of short-range resection in G_2_-phase DNA-PKcs deficient cells, we inquired about factors involved in long-range resection, reasoning that this stage is essential for hyper-resection [24,25,26]. Long-range resection is typically sustained by the activities of EXO1 or DNA2 in cooperation with BLM [61]. Appendix A shows the successful knockdown of EXO1 in parental and *DNA-PKcs* deficient A549 cells. However, the effect of this knockdown on resection is undetectable in both cell lines, thus ruling out EXO1 as a key factor of long-range resection in G_2_-phase (Appendix A).

Notably, individual knockdown of DNA2 or BLM (Figure 6A) suppresses almost completely resection in *DNA-PKcs* proficient, as well as in *DNA-PKcs* deficient cells and combined knockdown confers only limited additional effect (Figure 6B,C). To confirm this observation using alternative approaches, we tested a newly developed specific inhibitor of DNA2 [62]. Figure 6D,E confirm that DNA2 is the main contributor to long-range resection in G_2_-phase, independently of DNA-PKcs status, as DNA2i fully suppresses resection. We surmise that the hyper-resection detected in G_2_-irradiated *DNA-PKcs* deficient cells is caused by the unabated function of the canonical resection machinery rather than by the de novo engagement of alternative mechanisms.

## 4. Discussion

DNA-PKcs is a 469-kDa protein composed of 4128 amino acids, which makes it one of the largest kinases in higher eukaryotes. Significantly for this work, it is also one of the most abundant kinases in higher eukaryotes, particularly humans [11,33]. DNA-PKcs, when present in the nucleus contributes to repair and the regulation of transcription, but in the cytoplasm [63], it also has functions in energy metabolism [64], fatty acid synthesis [65] and aging [66]. Although the functions of DNA-PKcs continue to expand [67], its involvement in c-NHEJ has received the most attention and has been profoundly enriched by recent structural studies revealing intriguing aspects of its function in this repair pathway [68,69]. The work presented here helps to further develop and consolidate the functions of DNA-PKcs related to resection. Because resection is commonly regarded as the antipode to DSB processing by c-NHEJ, it also contributes to the development of organizing principles for DSB repair and DDR in general. 

### 4.1. Genetic Ablation of DNA-PKcs Causes Hyper-Resection in Cells Irradiated in G_2_-Phase

The application of resection-analysis methodology specifically in the G_2_-phase-irradiated cells allows us here to uncover DNA-PKcs functions in the regulation of resection in human and rodent cells. We find that as previously reported for *DNA-PKcs* deficient M059J cells and HCT116 *DNA-PKcs^−/−^* cells [27,28], also A549 *DNA-PKcs^−/−^* cells hyper-resect, as compared to their *DNA-PKcs* proficient counterparts. Interestingly, we show that this phenotype extends to rodent cells, despite their significantly lower constitutive levels of DNA-PKcs and three of the most widely used CHO *DNA-PKcs* mutants, V3, XR-C1-3 and irs20, hyper-resect, as compared to parental cells. Notably, this effect is detectable in all of these mutant cell lines both at low IR doses using QIBC as well as at high IR doses using FC.

The hyper-resection detected after DNA-PKcs-ablation may reflect a specific regulatory function of DNA-PKcs but may also result from the associated inhibition of c-NHEJ that non-specifically favors resection and thus HR. To distinguish between these two mechanisms, we analyzed resection in mutants with defects in c-NHEJ-factors other than *DNA-PKcs*. Strikingly, human cells with defects in *LIG4*, *XLF*, *PAXX* and *Artemis*, as well as rodent cells with defects in *Ku80* and *XRCC4,* fail to hyper-resect and show instead resection indistinguishable from their *wt* counterparts. Moreover, Ku knockdown using RNA interference in human or rodent cells fails to unleash hyper-resection, and notably, normal levels of resection are also measured after inhibition of DNA-PKcs enzymatic activity using a small-molecule inhibitor.

The hyper-resection uniformly observed in DNA-PKcs mutants by specific analysis in the G_2_-phase of the cell cycle is independently corroborated by the explosion in chromosomal translocation formation detected at metaphase when cells irradiated in the G_2_-phase are again specifically analyzed. While translocation formation increases only twofold in Ku80 and *XRCC4* CHO mutants as a result of their c-NHEJ defect (Figure 4E), an over five-fold increase is seen in a *DNA-PKcs* deficient mutant, pointing again to effects beyond the repair defect. Indeed, several studies link resection to translocation formation [47,48,55,56,57].

Because many of the c-NHEJ mutants examined in the above experiments show repair defects significantly stronger than *DNA-PKcs* deficient cells, our observations rule out general inhibition of c-NHEJ as the cause of hyper-resection. Indirectly, our results also rule out competition between c-NHEJ and HR as the basis for repair pathway choice (Figure 7A) [32,33,39,40]. They help to consolidate the hypothesis that hyper-resection is directly and specifically linked to DNA-PKcs ablation. We conclude that DNA-PKcs has a unique contribution to resection such that its ablation leads to hyper-resection (Figure 7B).

### 4.2. Integration of DNA-PKcs Functions with ATM and ATR Functions

The implication of DNA-PKcs in the regulation of resection generates parallels to the known effects of ATM and ATR on the same endpoint ([27,28] and references therein). This observation and previous results implicating DNA-PKcs also in G_2_-checkpoint recovery, allow us to further develop our model of DNA-PKcs, ATM and ATR integration to a module, the DNA-PKcs/ATM/ATR DSB-sensing module, in which the three kinases coordinately direct and organize DSB processing and DDR, presumably by strong crosstalk, guided by external chromatin cues.

A key and relevant characteristic of this module’s function is that interactions among participating kinases and the outputs to specific DSB repair pathways are strongly IR-dose dependent. At low IR doses (<2 Gy), resection (the key parameter to repair pathway choice) and the G_2_-checkpoint (a key component of DDR), are regulated by ATM and ATR in an integrated and epistatic manner, with output to the cell cycle only by ATR through CHK1. The integration of DNA-PKcs to the regulation of resection allows us to postulate that for DSBs undergoing resection under these conditions, DNA-PKcs act as a mediator to the engagement of ATM and ATR in their processing. Thus, for DSB-subsets responding to IR-dose (see below), the repair will be guided by the coordinated function of the DNA-PKcs/ATM/ATR module, which holds and shepherds DSBs through the different stages of processing, while adapting accordingly to DDR outputs. For those DSBs, DNA-PKcs likely play a decisive role in repair pathway choice, shunting DSBs away from c-NHEJ and towards a resection-dependent repair pathway–normally HR.

The functional integration of DNA-PKcs, ATM and ATR detected at low doses of IR changes profoundly as the IR dose increases. Now, DNA-PKcs become more dedicated to c-NHEJ and ATM/ATR while remaining partly coupled, are also developing independent outputs to diverse DDR endpoints [27,28]. It is interesting, however, that despite evident changes in the type of responses elicited by DNA-PKcs/ATM/ATR at high versus low IR doses, the effect of DNA-PKcs ablation on resection shows no detectable changes, suggesting that DNA-PKcs may somehow orchestrate the underlying dose-dependencies.

### 4.3. DNA-PKcs Is a Candidate Regulator of HR with Decreasing IR Dose

The implication of DNA-PKcs in resection is significant because it helps to develop mechanistic models explaining the increased engagement of HR with decreasing IR dose [19]. Indeed, engagement of HR to nearly 50% of DSBs at IR doses below 0.5 Gy and the complete dependence of resection and G_2_-checkpoint on ATR [27,28], generate a conundrum because the underpinning processes take place on DSBs that are likely already captured by the DNA-PK holoenzyme. The results presented here resolve the conundrum by integrating DNA-PKcs into the regulation of resection. We speculate that DNA-PKcs and other cues that remain to be characterized are essential in the overall organization of DDR with decreasing IR dose. We also speculate in the model outlined below that DNA-PKcs integrates itself into the mechanism that rapidly adjusts HR contribution from 10% at 2 Gy to nearly 50% below 0.5 Gy.

### 4.4. A model of DNA-PKcs Function in DSB Repair Pathway Choice at Low DSB Loads

Our study complements work from other laboratories on the putative functions of DNA-PKcs in resection and the regulation of HR. The Meek laboratory used a comprehensive mutational approach to address whether DNA-PK affects HR [32,33]. They report that DNA-PK’s ability to affect HR is titratable, dependent on its enzymatic activity and regulated by extensive phosphorylations on 40 sites or more. Because the phosphorylations have pleiotropic effects, with some inhibiting c-NHEJ, while others promote HR while inhibiting c-NHEJ [70], the authors propose that DNA-PKcs is a key determinant in the choice between c-NHEJ and HR during DSB repair [32,33].

More recently, the Paull laboratory used a powerful biochemical approach to address whether the choice between c-NHEJ and HR is driven by competition between DNA-PK and the MRN complex [39,40]. They report that the activity of the MRN complex for processing and resection of DNA ends is dependent on DNA-PK and phosphorylated CtIP, thus ruling out the competition model, as our results do as well. This work is significant because it directly integrates DNA-PK into the activation of MRN-mediated DNA end processing.

In the above studies, as well as in the analysis of recently published structural results [68,69], it is always assumed that unleashing of DNA end processing requires the removal of DNA-PKcs from DNA ends. In our effort to integrate the above observations on activation of the resection machinery by DNA-PKcs and the differential role of DNA-PKcs phosphorylation on HR, we developed the following speculative model. We hypothesize that DNA-PKcs assumes a unique role in the regulation of DSB repair in the low dose region, where HR dominates, by remaining at the DSB site and possibly translocating inwards while separating from Ku (Figure 7C) [71] to promote resection. It is likely that in this constellation, DNA-PKcs can assume novel regulatory functions that go beyond facilitating DNA-end synapsis and regulating the activities of components of the c-NHEJ apparatus [68,69,72]. In this function, the modifications associated with the well-studied ability of DNA-PKcs to dissociate from DNA ends may also facilitate such inward translocation.

This model explains several observations that remain puzzling otherwise. Thus, DNA-PKcs can be detected long times after IR as a focus at DSB sites, which, of course, requires many more molecules on the site than those actually required for c-NHEJ. It also explains our results that Ku ablation fails to cause hyper-resection, as does inhibition of DNA-PKcs using inhibitors. Indeed, when DNA-PKcs is inhibited using small molecule inhibitors, it remains trapped in the DNA ends [8,46]—a property that was actually exploited in one of the above studies [40]. Moreover, Ku may separate from DNA-PKcs as DNA-PKcs translocates inwards away from the break to activate resection, and Ku may actually fall off the end once the DNA has become single-stranded.

## 5. Outlook

The implication of DNA-PKcs in the increased engagement of HR at low DSB loads also allows speculation as to why DNA-PKcs is so abundant in higher eukaryotes and particularly humans (~500,000 molecules per cell). If DNA-PKcs is central to repair pathway choice and contributes to the rise of HR at low doses of IR, even distribution across the genome at ~10 kb distances, may provide surveillance, improving genomic stability. Such a benefit is partly lost if DNA-PKcs were only involved in c-NHEJ—an error-prone repair pathway. It may also be relevant in this regard that the lifespan of organisms correlates well with levels of DNA end-binding activity [73].

## Figures and Tables

**Figure 1 cells-11-02099-f001:**
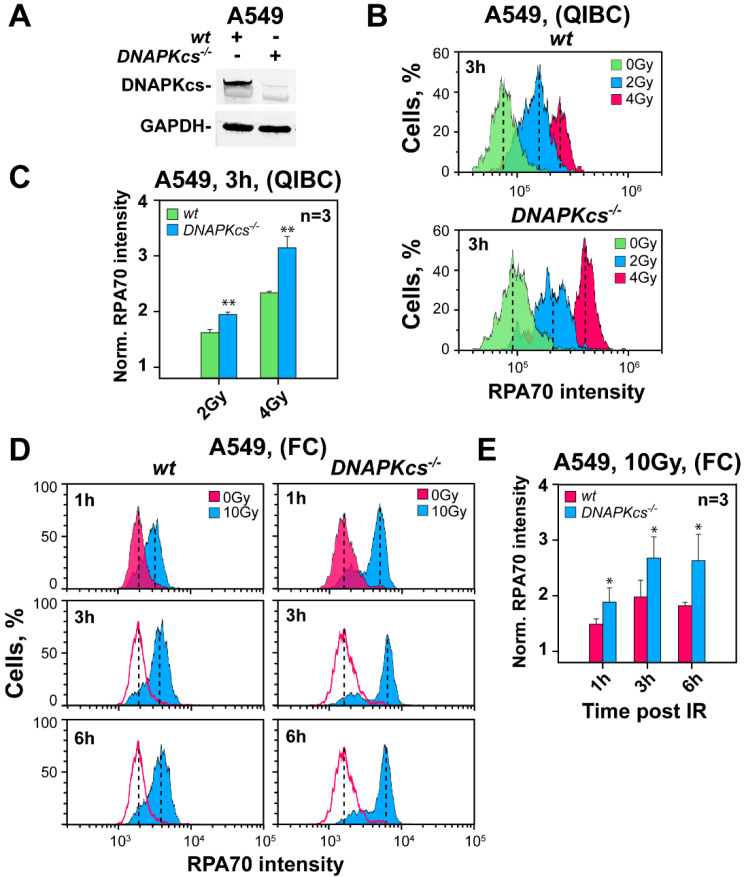
Detection in G_2_-phase of DNA end-resection in parental (*wt*) and *DNA-PKcs^−/−^* A549 cells. (**A**) Western blot showing strongly reduced DNA-PKcs levels in the selected clone of *DNA-PKcs^−/−^* A549 cells. (**B**) QIBC analysis of RPA70 signal at DSBs in *wt* and *DNA-PKcs^−/−^* A549 cells, 3 h after exposure to 2 Gy or 4 Gy. Resection is measured by calculating the arithmetic mean intensity of RPA70 signal in the histograms of G_2_-phase-irradiated cells (EdU^−^, G_2_-cells) and normalizing to the value measured in unirradiated controls (see Appendix A for details). (**C**) Bar plot showing the normalized RPA70 signal intensity from three experiments, obtained as shown in panel (**B**). (**D**) FC-based analysis of resection in *wt* and *DNA-PKcs^−/−^* A549 cells. Resection is measured 1, 3 and 6 h after exposure to 10 Gy (other details as in panel (**B**)). (**E**) Bar plot showing the normalized RPA70 signal intensity from three experiments, obtained as shown in panel (**D**). * indicates *p* < 0.05, and ** indicates *p* < 0.01.

**Figure 2 cells-11-02099-f002:**
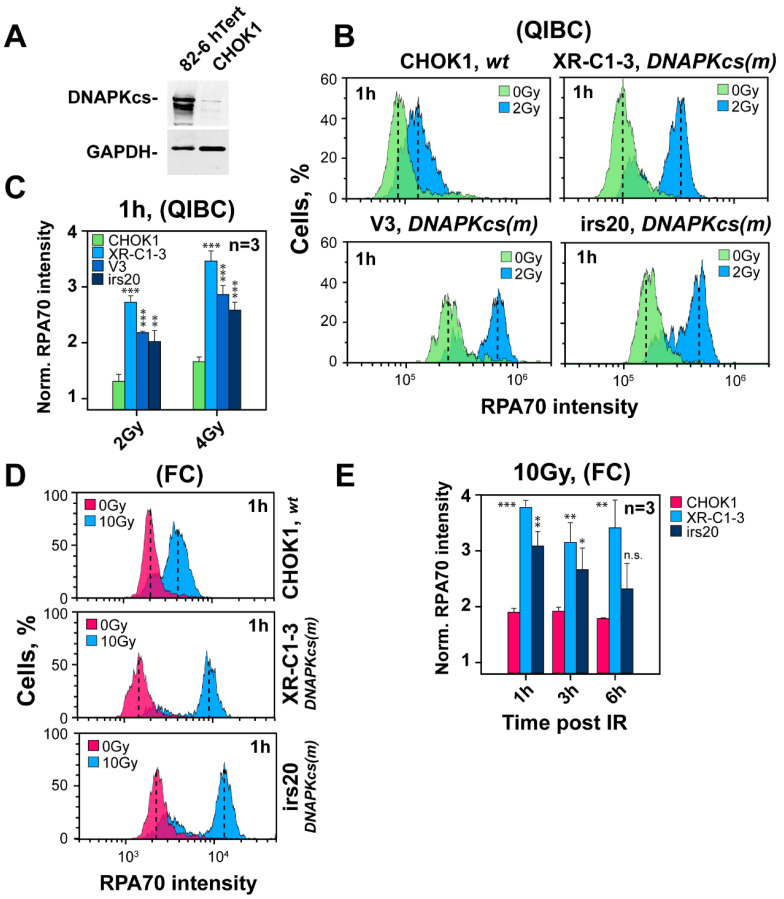
DNA-PKcs deficient CHO mutants uniformly show increased resection in G_2_-phase. (**A**) Western blot of DNA-PKcs in human 82-6 hTert and hamster CHOK1 cells. (**B**) As in Figure 1B for CHOK1 (*wt*) and the DNA-PKcs mutants, XR-C1-3, V3 and irs20. (**C**) As in Figure 1C for the mutants in panel (**B**). (**D**) As in Figure 1D for the mutants in panel (**B**). (**E**) As in panel (**C**) for data obtained using FC with the same mutants. * indicates *p* < 0.05, ** indicates *p* < 0.01, and *** indicates *p* < 0.001, while n.s. indicates non-significance.

**Figure 3 cells-11-02099-f003:**
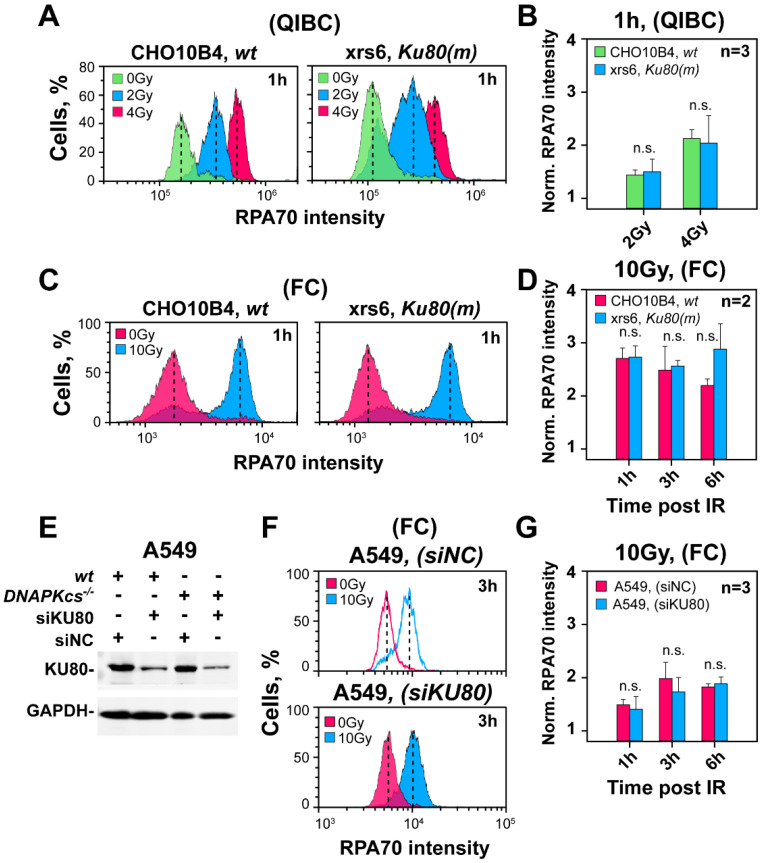
Ku deficiency leaves unchanged resection in G_2_-phase. (**A**) As in Figure 1B in CHO10B4(*wt*) and the Ku80-deficient mutant, xrs6. (**B**) As in Figure 1C for the cell lines in panel (**A**). (**C**) As in panel (**A**), following analysis by FC. (**D**) Quantitative analysis of results shown in panel (**C**). (**E**) Western blot showing the levels of Ku in A549 cells transfected with a siRNA targeting Ku80, or a non-specific control siRNA. (**F**) As in panel (**C**) for A549 cells transfected with a Ku80 specific siRNA, or a non-specific control siRNA. (**G**) As in panel (**D**) for the conditions shown in panel (**F**). n.s. indicates non-significance.

**Figure 4 cells-11-02099-f004:**
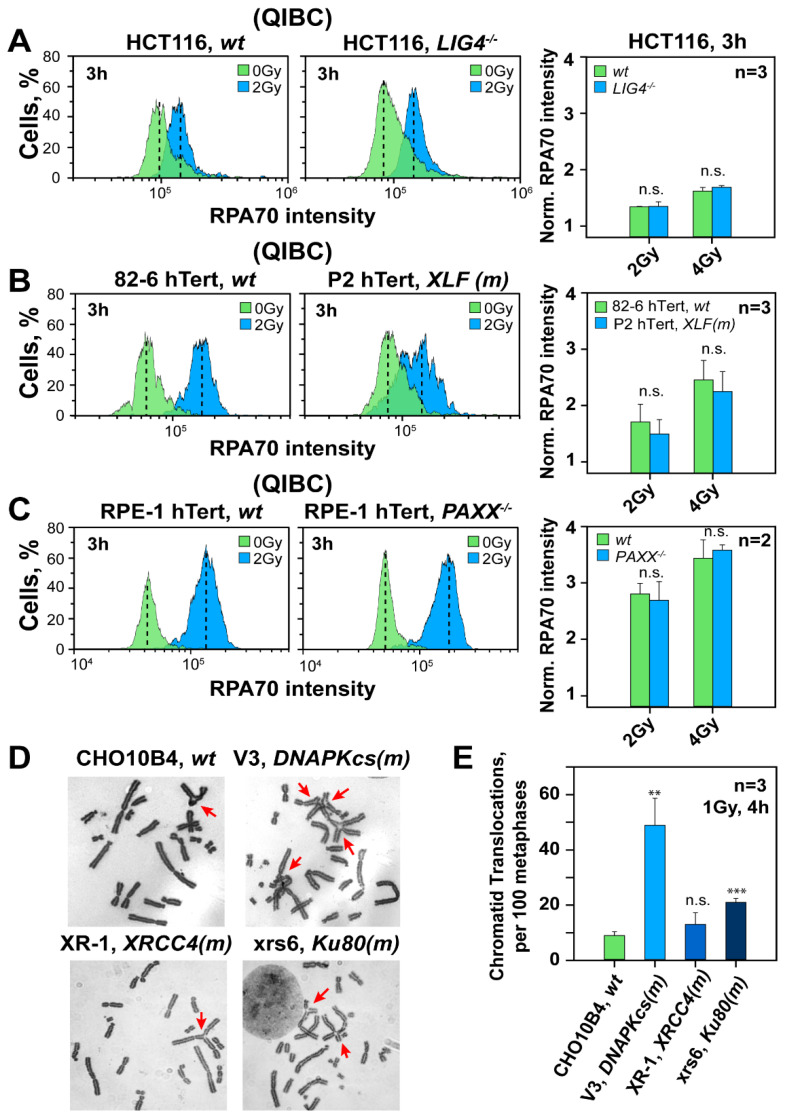
Deficiency in c-NHEJ factors other than DNA-PKcs leaves unchanged resection in G_2_-phase, but DNA-PKcs deficiency dramatically increases translocations. (**A**) QIBC analysis of RPA70 in parental and *LIG4^−/−^* HCT116 human cells in G2-phase. Bar plots on the right represent the normalized RPA70 signal intensity from three experiments. (**B**) As in panel (**A**) for 82-6 hTert and the P2 hTert fibroblast cell line with a defect in *XLF*. (**C**) As in panel (**A**), for RPE-1 hTert cells and a derivative *PAXX^−/−^* cell line. (**D**) Representative metaphase spreads of CHO10B4, V3, XR-1 and xrs6 cells depicting chromatid translocations at 4 h post 1 Gy IR. Translocations are marked with arrows (**E**) Frequency of IR induced translocations in *wt* CHO10B4 and c-NHEJ mutants (V3, XR-1 and xrs6 cell lines) at 4 h post 1 Gy IR. The mean ± SD shown represent data from three independent experiments. ** indicates *p* < 0.01 and *** indicates *p* < 0.001, while n.s. indicates non-significance.

**Figure 5 cells-11-02099-f005:**
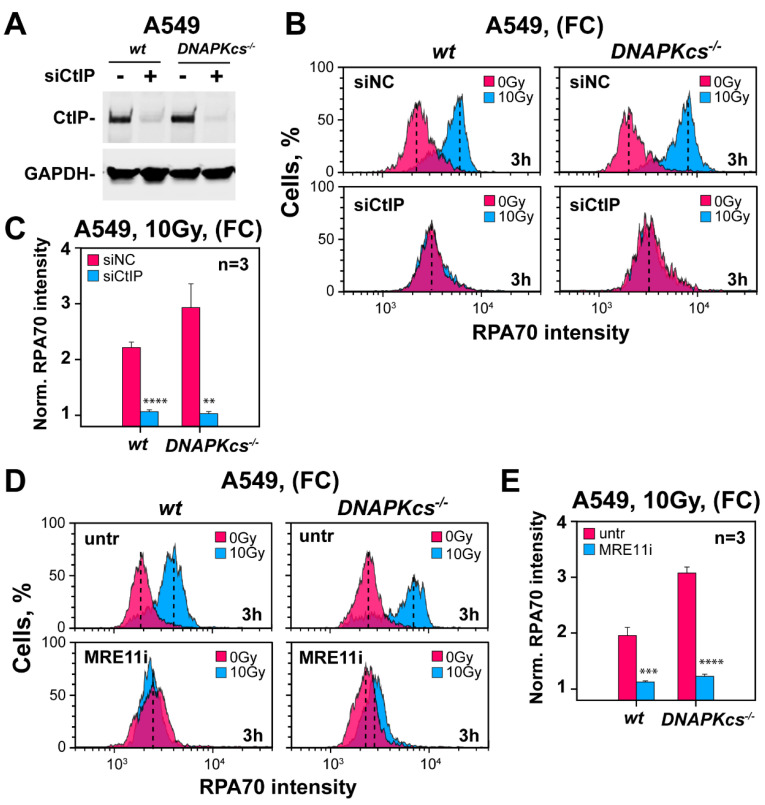
Resection in DNA-PKcs deficient and wild-type cells is entirely dependent on CtIP and MRE11. (**A**) Western blot analysis of CtIP levels in A549 wt and DNA-PKcs^−/−^ cells after transfection with a siRNA targeting CtIP, or a non-specific control siRNA. (**B**) As in Figure 1D for A549 *wt* and *DNA-PKcs^−/−^* cells following CtIP depletion and exposure to 10 Gy of IR. (**C**) As in Figure 1E for the results shown in panel (**B**). (**D**) As in panel (**B**), for cells treated with a cocktail of the MRE11 inhibitors mirin and PFM01. (**E**) As in panel (**C**) for the results shown in panel (**D**). ** indicates *p* < 0.01, *** indicates *p* < 0.001 and **** indicates *p* < 0.0001.

**Figure 6 cells-11-02099-f006:**
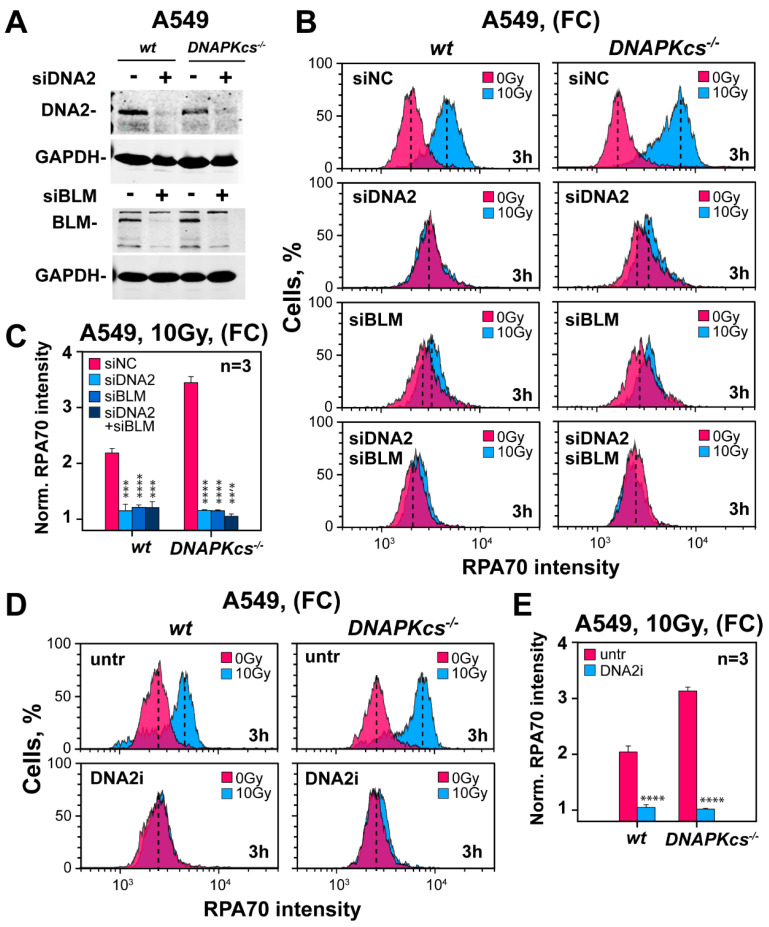
Depletion of DNA2/BLM nuclease-helicase complex suppresses resection in *wt* and *DNA-PKcs^−/−^* A549 cells. (**A**) Western blot analysis showing levels of DNA2 and BLM proteins after transfection of parental and *DNA-PKcs^−/−^* A549 cells with siRNAs targeting these proteins, or with a non-specific control siRNA. (**B**) As in Figure 1B for the conditions shown in panel (**A**) in cells exposed to 10 Gy of IR. (**C**) As in Figure 1C for the results in panel B. (**D**) As in panel (**B**), for A549 cells treated with 4 µM of a specific DNA2 inhibitor, (DNA2i). (**E**) As in panel (**C**) for the results in panel (**D**). *** indicates *p* < 0.001 and **** indicates *p* < 0.0001.

**Figure 7 cells-11-02099-f007:**
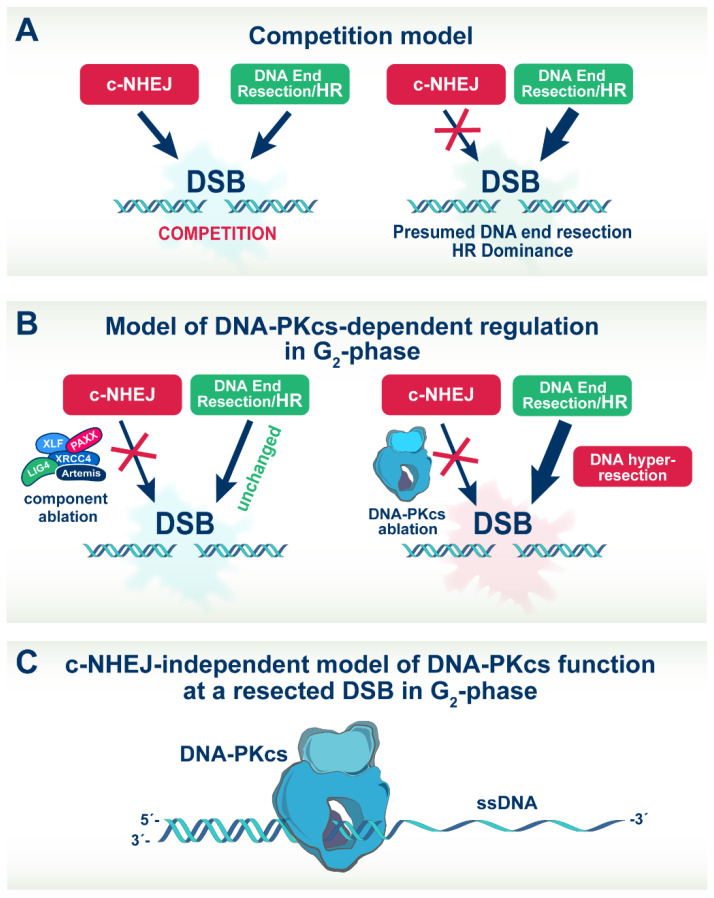
Models of DNA-PKcs function in DSBs repair pathway choice. Model (**A**): It is commonly assumed that choice between c-NHEJ and HR is the outcome of competition at the DSBs, at a first-come first-serve basis; Model (**B**): Our results allow us to modify model A by showing that DNA-PKcs uniquely contributes to resection, in ways that are not shared by other c-NHEJ factors, its ablation causes, therefore, hyper-resection; Model (**C**): A speculative function of DNA-PKcs in the regulation of DSB repair. Our results are compatible with DNA-PKcs remaining at the DSB site, possibly translocating inwards to promote resection. This model might be more relevant for low DSBs loads. For details see text.

## Data Availability

Appendix A are available in the Appendix A accompanying this manuscript.

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
