# Peer review of "Increased Resection at DSBs in G2-Phase Is a Unique Phenotype Associated with DNA-PKcs Defects That Is Not Shared by Other Factors of c-NHEJ"

_cells, 2022, doi:10.3390/cells11132099_

Round 1

Reviewer 1 Report

The manuscript presented by Xiao et al. focuses on the competition between 2 molecular pathways of DNA repair, particularly cNHEJ and HR. This topic is very interesting and their findings will sure make a significant contribution to the general knowledge of DNA repair dynamics in mammals. It is well written with satisfactory experimental design. Nevertheless, I would really strongly recommend authors to make the data presentation a bit easier by providing the microscopy image along with the charts and some summary graphical abstract or similar.

Specific Comments:

Though I believe that the QIBC is a very powerful tool,  find the very many FC pictures rather confusing, particularly because it is hard to imagine how the histograms were generated. The Fig. S1 which is supposed to explain the method is not clear, e.g.:

a) what is the red line in the RPA70 chart

b) It is not clear the EdU  positive and EdU negative gating, perhaps different color would help

c) please show how the RPA70 signal was gated and collected

 d) Why the arithmetic average was used instead of  the peak area 

In the discussion authors mention that DNA PKC depletion migh cause the inhibition of c-NHEJ, and via this way stimulate the HR. Depletion of individual factors of cNHEJ, however, has no effect. I wonder how authors explain this.

a) Would the simultaneous depletion of KU and Ligase show clearer results and  is the Ku80/70 complex fully dysfunctional when Ku80siRNA is used? Do authors consider activation of alternative NHEJ?

b) Isnt the DNAPKC function more related to the blocking the over-action of MRN rather than in  inhibitting the cNHEJ? I am not an expert on the DNA PKC - is its interactome known?, e.g. does it interact directly with the NHEJ and HR components? It could also form a physical bareer around the break which endonucleases cannot overcome.  

Minor points:

line 277: original reference for the resection measurement is missing

Full names of the proteins are usually given at first mention (ATM, ATR, BLM..etc)

Author Response

Reviewer 1

General Comments

“The manuscript presented by Xiao et al. focuses on the competition between 2 molecular pathways of DNA repair, particularly c-NHEJ and HR. This topic is very interesting and their findings will sure make a significant contribution to the general knowledge of DNA repair dynamics in mammals. It is well written with satisfactory experimental design. Nevertheless, I would really strongly recommend authors to make the data presentation a bit easier by providing the microscopy image along with the charts and some summary graphical abstract or similar.”

We thank the Reviewer for the positive evaluation of our work. We address below some of the more specific comments formulated here.

Specific Comments:

“1. Though I believe that the QIBC is a very powerful tool, find the very many FC pictures rather confusing, particularly because it is hard to imagine how the histograms were generated. The Fig. S1 which is supposed to explain the method is not clear, e.g.:a) what is the red line in the RPA70 chart b) It is not clear the EdU  positive and EdU negative gating, perhaps different color would help c) please show how the RPA70 signal was gated and collected  d) Why the arithmetic average was used instead of the peak area” 

We thank the Reviewer for pointing out this shortcoming in the description of a key method. In the revised manuscript we include an updated version of Figure S1, which better illustrates the approach used to define gates. We have also updated the legend of Figure S1 and have modified the corresponding section of “Materials and Methods” to include detailed outline of QIBC, and the analysis of generated data. More specifically:

  1. The red line in Figure S1 (now substituted by a black line), shows the arithmetic mean and is drawn to facilitate comparisons of RPA70 signal among samples.
  2. We updated the figure to indicate EdU-positive and EdU-negative cells with different colors.
  3. The updated version of Figure S1, illustrates better the EdU-based gating of the RPA70 signal.
  4. Arithmetic mean and Median mean are equivalent ways of data analysis. Choice of either parameter has no effect on the appearance or interpretation of results. Our choice of Arithmetic mean is arbitrary, but has been used in the majority of our publication thus far. Therefore, we retain this way of presentation in the present paper as well.

“2. In the discussion authors mention that DNA PKC depletion might cause the inhibition of c-NHEJ, and via this way stimulate the HR. Depletion of individual factors of cNHEJ, however, has no effect. I wonder how authors explain this.”

This conundrum is central to the design of the work presented in the present paper. The fact that we can now be sure of the unique role of DNA-PKcs in the investigated responses, will allow us to embark with confidence into studies of the underpinning mechanisms. Our best-guess explanations thus far are outlined in the Discussion.

  1. Would the simultaneous depletion of KU and Ligase show clearer results and is the Ku80/70 complex fully dysfunctional when KU80 siRNA is used? Do authors consider activation of alternative NHEJ?

Our data show that DNA-PKcs regulates resection actively, rather than through inhibition of c-NHEJ passively. In addition, we did not find an effect after KU or Ligase 4 depletion. Therefore, we do not anticipate different responses following combined depletion. At the level of Ku depletion achieved in the experiment shown, we cannot exclude residual activity. This is why we address this question using different cell lines and from different perspectives. Effects on resection will affect alt-EJ [1]. However, these responses are not part of the present work and will be studied in more detail in future work.

  1. Isnt the DNAPKC function more related to the blocking the over-action of MRN rather than in  inhibitting the cNHEJ? I am not an expert on the DNA PKC - is its interactome known?, e.g. does it interact directly with the NHEJ and HR components? It could also form a physical bareer around the break which endonucleases cannot overcome.  

This is a very good point. We summarize in the Introduction and Discussion existing knowledge in the field on this topic and connect it to our observations.

Minor points:

“1. line 277: original reference for the resection measurement is missing.”

We added the reference.

2. Full names of the proteins are usually given at first mention (ATM, ATR, BLM..etc).”

We have defined acronyms at first mention.

  1. Howard, S.M.; Yanez, D.A.; Stark, J.M. DNA Damage Response Factors from Diverse Pathways, Including DNA Crosslink Repair, Mediate Alternative End Joining. PLoS Genetics 2015, 11, e1004943, doi:10.1371/journal.pgen.1004943.

Reviewer 2 Report

Xiao et al present a study investigating the link between DNAPKcs and end resection after treatment with ionizing radiation (IR). They use multiple lines of evidence to demonstrate hyperresection in G2 cell cycle phase at doses of 2-10 Gy. They show that depletion of c-NHEJ factors does not have the same effect, ruling out that this is the result of a simple competition between HR and c-NHEJ, and that resection is dependent on CtIP and MRE11 showing that it is true end resection. There is a growing interest in targeting DNAPKcs pharmacologically in cancer, and thus defining the very complex role of this protein is of great importance. The manuscript is timely, and will likely be of great interest to the readers of cells.

Overall, the manuscript is well written, and the figures are clear, well-labelled, and concise. I have some concerns and comments that should be addressed to improve the manuscript.

1) The observations made in this manuscript measure resection at higher radiation doses, at or above 2 Gy where 10% or less of the DSBs will be repaired by HR. Yet, the authors extrapolate their mechanistic insights to this low dose regime and HR. How can the authors show that the resection events observed are leading to HR vs alternative pathways. Showing that DNAPKcs also regulates resection at lower doses (0.5 Gy) where HR is dominant would be important.

2) The manuscript would benefit greatly from a summary figure or supplementary figure that summarized their proposed model for the function of DNAPKcs in HR.

3) QIBC analysis histogram plots (Figure 1B, 3A) the three doses (0 Gy, 2 Gy, and 4 Gy) should be shown in a single plot to facilitate the comparison between the three doses, rather than as two separate plots.

4) The mean fluorescent intensity (MFI) is often used for the expression of target protein across per cell. However, the data in 3A and 3C showed a positive population shift through the percentage of gated cells. How would the authors justify the EDU- population shift towards positive RPA70 in ku80 mutant CHO cells after irradiation (Figures 3A and 3C)?

Author Response

Reviewer 2

General Comments

“Xiao et al present a study investigating the link between DNAPKcs and end resection after treatment with ionizing radiation (IR). They use multiple lines of evidence to demonstrate hyperresection in G2 cell cycle phase at doses of 2-10 Gy. They show that depletion of c-NHEJ factors does not have the same effect, ruling out that this is the result of a simple competition between HR and c-NHEJ, and that resection is dependent on CtIP and MRE11 showing that it is true end resection. There is a growing interest in targeting DNAPKcs pharmacologically in cancer, and thus defining the very complex role of this protein is of great importance. The manuscript is timely, and will likely be of great interest to the readers of cells. Overall, the manuscript is well written, and the figures are clear, well-labelled, and concise. I have some concerns and comments that should be addressed to improve the manuscript.”

We thank the Reviewer for his positive comments on the manuscript.

Specific Comments

“1. The observations made in this manuscript measure resection at higher radiation doses, at or above 2 Gy where 10% or less of the DSBs will be repaired by HR. Yet, the authors extrapolate their mechanistic insights to this low dose regime and HR. How can the authors show that the resection events observed are leading to HR vs alternative pathways. Showing that DNAPKcs also regulates resection at lower doses (0.5 Gy) where HR is dominant would be important.”

The Reviewer makes a valid point indeed. However, measuring RPA signal at doses well-below 1 Gy is challenging, even with QIBC. We are currently working on improvements to enhance sensitivity and possibly enable such measurements. However, because the effects on resection observed in this paper fail to show the strong dependence on IR dose observed in our earlier work for RAD51, lack of RPA results below 1-2 Gy is not in any way limiting the power of our conclusions.

“2. The manuscript would benefit greatly from a summary figure or supplementary figure that summarized their proposed model for the function of DNAPKcs in HR.”

Thanks a lot for the suggestion. We have added a summary figure.

“3. QIBC analysis histogram plots (Figure 1B, 3A) the three doses (0 Gy, 2 Gy, and 4 Gy) should be shown in a single plot to facilitate the comparison between the three doses, rather than as two separate plots.”  

Excellent suggestion. In the revised version of the manuscript the indicated figures are shown combined.

“4. The mean fluorescent intensity (MFI) is often used for the expression of target protein across per cell. However, the data in 3A and 3C showed a positive population shift through the percentage of gated cells. How would the authors justify the EDU- population shift towards positive RPA70 in Ku80 mutant CHO cells after irradiation (Figures 3A and 3C)?”

We believe that this effect reflects peculiarities of the protocol used to detect RPA. As we describe under “Material and Methods”, unbound RPA is extracted during sample preparation by incubating cells in PBS containing 0.25% Triton X-100. In this way, we actually measure only bound and not total RPA levels which are expected to increase when resection occurs. Therefore, IR induces in EdU- G2 cells an increase in RPA signal. Changes in base-line in the KU80 mutant may therefore reflect change in chromatin that alter this base line. This is the reason why we only use differences (irradiated versus non-irradiated) of wild type cells and the mutant to draw conclusions.